# The Accelerated Right Ventricular Failure in Fetal Anemia in the Presence of Restrictive Foramen Ovale

**DOI:** 10.3390/diagnostics12071646

**Published:** 2022-07-06

**Authors:** Suchaya Luewan, Fuanglada Tongprasert, Kasemsri Srisupundit, Theera Tongsong

**Affiliations:** Department of Obstetrics and Gynecology, Faculty of Medicine, Chiang Mai University, Chiang Mai 50200, Thailand; suchaya.l@cmu.ac.th (S.L.); jeab094@hotmail.com (F.T.); kasemsri.s@cmu.ac.th (K.S.)

**Keywords:** anemia, fetal bleeding, heart failure, restrictive foramen ovale

## Abstract

***Objective:*** To describe serious hemodynamic changes secondary to anemia in the case of restrictive foramen ovale (FO). ***Case:*** A 43-year-old pregnant woman, G4P0030, underwent fetal echocardiography at 35 weeks of gestation and was found to have (1) restrictive FO; (2) poor right ventricular function; (3) unbalanced hemodynamics; (4) fetal anemia (high MCA-PSV and hepatosplenomegaly). Acid-elution test indicated feto-maternal hemorrhage. Cesarean section was performed for postnatal blood transfusion. Nevertheless, the newborn developed heart failure and died after partial blood exchanges. ***Conclusions:*** Insights gained from this study are as follows: (1) Restrictive FO in structurally normal hearts can modify fetal response to anemia differently, by unequally distributing blood volume, leading to much more deteriorating right ventricular function. (2) To make decisions for intrauterine or extrauterine treatment in cases of anemia-associated heart failure, several factors must be taken into account such as gestational age, fetal cardiac function, and placental function. Because of the hyperdynamic state of newborns immediately after birth, delivery can deteriorate the compromised heart to irreversible failure. Intrauterine transfusion for a well-prepared heart just before delivery may be the best option since the baby should be well oxygenated at the time of delivery.

## 1. Introduction

Restrictive foramen ovale (FO) is often associated with left side obstruction (mitral stenosis, aortic stenosis and hypoplastic left heart syndrome), resulting in increased pressure in the left atrium and back pressure on the flap of FO bulging back to the right atrium. Restrictive FO in cases of structurally normal hearts is usually considered as a normal variation with good outcomes [1], though it can cause less circulation in the left side and left-right disproportion. Fetal echocardiography may show proportional cardiac chambers and a restrictive ostium secundum atrial septal defect, with no other abnormalities. The flap of FO is usually redundant or fixed bulging in the left atrium, also called atrial septal aneurysm (ASA). Several cases of restrictive FO or ASA have been prenatally diagnosed [2,3,4,5,6]. It is frequently described as an uncommon finding, although in one large study a prevalence of 7.6% was reported [7]. Additionally, several studies report a relationship between ASA and arrhythmias and some reports found a high prevalence of ASA, as many as 70%, among fetuses referred for evaluation of arrhythmias [8,9]. ASA usually is not a pathological condition, being self-limited, with some authors defending that it should be considered benign and transient [7]. After delivery, in most of the cases, because of the physiological vascular modifications, there is a normalisation of the ventricular size and a repositioning of the atrial septum [10]. Though ASA is usually benign or a functional change, it should be considered and excluded due to its implications in prenatal diagnosis counselling, as it can mimic a left side structural heart disease [11]. Additionally, in cases where there is a severe bulging of the redundant septum primum into the left atrium, it can compromise the flow through the mitral valve, creating the appearance of ventricular disproportion, though the prognosis is usually good. Among fetuses with left-right disproportion, however, other causes of right ventricular failure must be further investigated, since the restrictive FO of the structurally normal heart is very unlikely to cause heart failure [1], though in rare cases, especially severe restrictive or close FO, it may be associated with right ventricular dilation, tricuspid regurgitation, pericardial effusion, and heart failure [12] Theoretically, in cases of increased need of blood flow through the FO to increase cardiac output as seen in fetal anemia, the restrictive FO may cause right heart failure. This study highlights the effect of anemia on restrictive FO, leading to heart failure. 

In general, fetuses can tolerate anemia well without heart failure, even after hydrops fetalis has already developed [13,14,15]. Fetal anemia is usually associated with increased cardiac size and blood volume to increase cardiac output for tissue oxygen perfusion. Typically, the heart is globally and equally enlarged both sides. Heart failure usually occurs long after development of hydrops fetalis, when the compensatory mechanism becomes exhaustive [14]. The objective of this report is to describe serious hemodynamic changes secondary to anemia in the case of restrictive FO in which the increased blood volume could not be equally distributed to both sides.

## 2. Case Presentation

This case study was carried out with ethical approval by the Institutional Review Boards (IRB) and the patient provided written informed consent.

A 43-year-old pregnant woman, G4P0030, presented with decreased fetal movements at gestational age of 35 weeks. The course of current pregnancy was uneventful. Ultrasound scans at mid-pregnancy were normal. Because of decreased fetal movement, a nonstress test was performed and was reactive. Detailed ultrasound revealed that the fetal heart was slightly enlarged (cardio-thoracic diameter ratio: 0.61), and structurally normal. However, with careful inspection, the ultrasound showed disproportion of the left and right side (atria, ventricles and great arteries). The right side was larger. Furthermore, the contractility of the right ventricle was poorer than that of the left side. The size of the right ventricle (RV) during systole and diastole was not very different, whereas the left ventricle (LV), though small in size, during systole was obviously collapsed (Figure 1). On M-mode, the RV showed poor contraction. Shortening fraction of the RV was 17% compared with 32% of the LV. The interventricular septum was bulged to the left side during systole, suggestive of pressure or volume load in the RV. Additionally, movement of the flap of foramen ovale (FFo) showed some restriction with a size of approximately 2.3 cm diameter, and was relatively fixed and had tense ballooning during both systole and diastole. Blood flow in the bulging flap partially crossed the foramen and partially turbulently flowed back to the right atrium (RA) (Figure 1). No sign of hydrops fetalis was observed. 

Fetal hemodynamics were comprehensively evaluated (Figure 2). Cardiac performance (Tei) index of the RV was markedly increased (0.8), indicating poor function. ICT (35 ms) and IRT (58 ms) were also prolonged, predominantly in ICT or systolic dysfunction, suggesting that volume load was more pronounced than pressure load. On the contrary, Tei index of the LV was normal (0.27) and shortening fraction was also normal (34%). Triscupid regurgitation (141 cm/s) was noted. Cardiac output of RV showed Z-score of 0.1 or normal (measured value 632 mL/min vs expected value 612.5 mL/min for 35 weeks of gestation). Preload index in the ductus venosus (S-a/S), representing central venous pressure, was significantly increased (measured value 0.774 vs expected value 0.491). Umbilical venous pulsations were also noted. LV and left outflow tract were small but patent with laminar flow entering the ascending aorta (AAo), with no evidence of flow obstruction. Pulsed Doppler of the pulmonary vein was normal with positive a-wave. Normal umbilical artery Doppler wave forms were demonstrated, indicating normal placental blood flow. The AoA was smaller than the ductal arch (DA) and there was reversed flow in the AoA. In the middle cerebral artery, peak systolic velocity (MCA-PSV) was assessed and was 93.99 cm/s (1.8 MoM), indicating fetal anemia. The liver and the spleen were slightly but significantly enlarged (measured liver length 5.92 cm; 97.5th percentile value: 5.27). Normal umbilical artery Doppler wave forms were demonstrated, indicating normal placental blood flow.

The positive findings show some degree of restrictive FFo, reversed flow in the aortic arch, poor RV function or impending failure (increased preload, increased Tei index, reduced shortening fraction, and relatively low cardiac output) and high velocity in the MCA, indicating fetal anemia [16,17]. The major concern of the findings is right ventricular failure. The hemodynamics of this case are unusual and the primary cause of poor cardiac function is yet to be elucidated. At this time, we do not know what is the cause of fetal anemia and the anemia is not typical, as seen in most cases in which the heart is globally enlarged, equally both sides. Nevertheless, the findings may be explained by restrictive FFo, resulting in right-left disproportion, which can modify the fetal response to anemia to be atypical, i.e., predominant right side enlargement, instead of global enlargement. The atypical findings may be summarized as follows: In anemia, the heart is usually enlarged globally and equally without disproportion [17].Reversed flow in aortic arch is commonly seen in hypoplastic left heart or aortic stenosis but in this case the flow crossing the aortic valve shows normal laminar flow.Reversed flow in aortic arch is different from that seen in fetal growth restriction (FGR), which usually shows increased afterload (abnormal umbilical blood flow).


The most likely explanation is that restrictive FO aggravated cardiac overload of the right side, leading to volume load to the right ventricle. Blood from the right ventricle runs through the pulmonary artery and the ductus arteriosus and to the placenta via descending aorta, typically joining blood from the aortic arch. In this case, blood volume in the left side is much less than the right side. Therefore, blood from the right side runs through the ductus and reversely runs in the aortic arch to feed the neck vessels. The flow reversal in the aortic arch was likely caused by the differential pressure in the ductus arteriosus and the aortic arch, which was contributed to by volume load in the right side. The phenomenon of reversed flow in the distal aortic arch secondary to fetal anemia has been described before [18,19]. This is likely associated with an increase in cerebral and coronary blood flow and vascularity due to low oxygen delivery, leading to a reduction in the peripheral resistance in both organs due to autoregulation. As a result, pressure difference and reversal of flow occur.

Differential diagnoses of fetal anemia: Hematologic disorders (alpha-thalassemia, red cell membrane defect, etc.), infection (syphilis, parvovirus B19 etc.), alloimmunization (Rh blood group), feto-maternal hemorrhage, etc.

Differential diagnoses of reversed flow in the AoA: (1) Aortic stenosis; (2) Hypoplastic left heart syndrome; (3) Increased afterload (resistance in the placenta) in case of fetal growth restriction; (4) Severe restriction of the FO. Reversed flow in the AoA seen in this case is an attempt to fill the left side in spite of low resistance in the placenta (normal end-diastolic flow in the umbilical artery) because of the two factors: (1) less volume secondary to restrictive FO, and (2) hypervolemia due to fetal anemia. 

A work-up to identify the causes of fetal anemia were performed. The cause was found to be feto-maternal bleeding (acid elution test showed a ratio of fetal to maternal red cells of 5:100, approximately 150 mL of fetal blood bleeding to maternal circulation) (Figure 3). Cesarean section was performed for neonatal treatment rather than intrauterine blood transfusion (IUT), giving birth to a male baby, weighing 2000 g, Apgar scores 6 to 6 to 8 (tube). Umbilical cord blood gas showed mild metabolic acidosis. Neonatal hemoglobin was 2.5 g/dL. Because of marked anemia, neonatal blood partial exchange was performed. However, heart failure progressively worsened and was intensively treated. Due to anemic hypoxia and compromised heart, the baby had cardiogenic shock with prolonged metabolic acidosis, refractory to treatment and ran down, in spite of intensive care. The baby died on day 2 of life. An autopsy was performed and confirmed the prenatal findings (the foramen size: 1.9 × 2.8 mm; left-right side disproportion and small foramen ovale) (Figure 4).

Obviously, fetal anemia was caused by massive bleeding. Fetal blood transfusion is the definitive treatment. However, experience of intrauterine treatment of heart failure is limited. Retrospectively, the decision on delivery for postnatal blood transfusion/exchange or intrauterine blood transfusion is very challenging in these cases. Though 35 weeks is very near term, it is preterm. Upon delivery, the baby takes on more burdens in having to adapt to several changes. Thus, fetal heart overload may be a consequence of birth itself because of temperature adaptation, fluid loss from skin of the premature baby, and transition from placental respiration to lung respiration requiring a well-oxygenated heart. At the first breath, after cutting the umbilical cord, the baby must cope with serious conditions which might increase cardiac load and aggravate the compromised heart. The baby should be well prepared for birth by IUT to have a well-oxygenated heart before birth rather than birth with anemic hypoxia followed by extrauterine treatment. The advantage of EUT is convenience for more effective procedures, whereas IUT is more complicated to perform, but more importantly is a definitive treatment in terms of getting rid of the cause (FMH) after cutting cord. 

## 3. Discussion

In general, fetuses can tolerate with anemia well without heart failure even after hydrops fetalis has already developed [13,14,15]. This case demonstrates that restrictive FO, considered as a normal variation with good outcomes [1], can modify the fetal response to anemia by aggravating the RV to failure because of unbalanced distribution of the increased blood volume secondary to anemia. Unlike heart failure caused by anemia, which is usually global failure on both sides, this case showed asymmetrical heart failure with preserved LV. In this case, fetal bleeding causes anemia and hypoxia, leading to hemodynamic adaptations to maintain peripheral oxygen perfusion by increasing cardiac output and blood volume. Typically, heart failure occurs long after development of hydrops, which is a consequence of hypervolemia, not heart failure [14]. This study underlines that the benign restrictive FO can modify fetal hemodynamics from a mild to life threatening condition.

**Importance of restrictive FO:** There is substantial evidence that primary closure of the FO results in small size of left heart structures [20,21], probably associated with less blood volume running through the left heart structures. The effect of unequal redistribution of blood flow, predominantly on the right side, on cardiac function in later life is unknown. Both studies in animal fetuses [22,23] and in human fetuses [24,25,26,27] have shown that several adaptive changes in FO blood flow volume occur during hypoxemia or hypovolemia, implying that quantifying FO blood flow and its redistribution may play a critical role in assessing fetal adaptation to oxygen insufficiency. With advances in ultrasound technology, it is currently feasible to assess FO blood flow volume with more reliability. It has been shown to be feasible in measurement of the vessel blood flow in fetal lambs [28] and human fetuses in late pregnancy [29,30,31,32], and has been used successfully to make initial observations of redistribution of the fetal hemodynamics in human fetuses with congenital heart disease [33,34]. The FO blood flow may be best estimated by pulsed Doppler ultrasound together with spatio-temporal image correlation, comparing the measured values with the reference ranges, from 20 to 40 weeks of gestation [35]. Restrictive FO is usually caused by an increase in pressure in the LA, especially associated with hypoplastic left heart, aortic stenosis or mitral valve stenosis. The flap of FO tends to bulge to the RA and be fixed or flat. On the contrary, restrictive FO with normal structures, like our case, is prenatally well tolerated, though the right side is overloaded due to difficulty in crossing through the FO. Most have redundant flaps of FO, ballooning to the LA [1]. Restrictive FO associated with left heart failure is commonly reported but that associated with right heart failure has never been described. 

**Significance of reversed flow in the AoA:** As mentioned earlier, this is usually associated with less blood flow in the left side, especially aortic stenosis. Consequently, blood flow from the RV has compensatory filling to the left side via ductus arteriosus and reversal in the AoA. Thus, reversed flow in the AoA usually implies left heart disorders. In some cases of ASA, both left-right disproportion with a predominant right side and retrograde flow in the aortic arch are identified. In such cases, differential diagnosis must include, mainly, causes of severe left side structural heart disease, such as hypoplastic left heart, aortic coarctation and aortic stenosis or aortic atresia [36]. In some cases, the reduced circulation in the left side can be significant to the point of causing a retrograde flow in the aortic arch, raising the suspicion of an important left side cardiac structural disease. Additionally, reversed flow in the AoA may be caused by increased afterload as seen in fetuses with growth restriction or placental insufficiency [37], as it is more difficult to circulate to the placenta. However, reversed flow in the AoA with a structurally normal heart as seen in this case is caused by an attempt to fill the left side because of volume load in the RV, despite low placental resistance (normal umbilical artery Doppler velocity). Accordingly, reversed flow in the AoA in this case is unique and is different from previous reports.

**Hemodynamic change in response to fetal anemia:** The fetus has very high capability of adaptation to anemia without cardiac compromise. Hydrops fetalis is caused by hypervolemia with high vascular permeability rather than heart failure since most cases of anemic hydrops fetalis have normal cardiac function without increased central venous pressure [13,14,15]. The most common sonographic findings, seen in early response to anemia, are increased cardiac size (cardiothoracic ratio), increased MCA-PSV (because of low viscosity), and hydrops fetalis (fluid collection in third spaces) without heart failure seen in the second phase (if anemia is corrected, cardiomegaly and hydropic signs can completely resolve). In fetal anemia, heart failure is a late consequence after longstanding uncorrected severely anemic hypoxia and usually develops long after the appearance of hydrops. Typically, cardiomegaly is symmetrical or globally enlarged with good function until late stage when compensatory mechanisms become exhausted on both sides. Interestingly, because of restrictive FO in this case, the burden of volume load from anemia is much more aggravated the RV than the LV. Accordingly, the overloaded heart and poor function was confined to the right side and more rapidly developed without hydropic changes. Normal shortening fraction and Tei index are maintained in the left side while the right ventricle is the first to fail (poor shortening fraction, a marked increase in Tei index, and increased preload). The restriction of FO prevented the LV from being overloaded but facilitated it to be even more progressively overloaded on the right side.

**What is the best management?** In therapeutic consideration, three possible challenging options are as follows: (1) intrauterine transfusion (IUT) and expectant management for delivery at term; (2) IUT and then urgent delivery; and (3) urgent delivery for post-natal treatment. In making a decision, several factors such as gestational age, fetal status, cardiac function etc., must be taken into account. The first option might be suitable for fetuses remote from term. For fetuses at term or near term like this case, delivery may be preferable since delivery is the definitive treatment for fetal bleeding, as there is no recurrence after postnatal blood transfusion. Postnatal treatment has advantages of less technical difficulty and more availability of hemodynamic assessment. However, immediately after delivery, the baby must adapt to several changes. Thus, fetal heart overload may be a consequence of birth itself because of temperature adaptation, fluid loss through skin of the premature baby, and transition from placental respiration to lung respiration, requiring a well-oxygenated heart. At the first breath, the baby must cope with serious conditions which might increase cardiac load and aggravate the compromised heart. The baby should be well prepared for birth by IUT to have a well-oxygenated heart before birth rather than birth with anemic hypoxia followed by extrauterine treatment.

**Lessons:** Retrospectively, this fetus should have been delivered in a well-prepared condition, since delivery is the definitive treatment of fetal hemorrhage and it was near term, presumably mature enough for the lung development. However, to mitigate the neonatal heart burden immediately after birth, fetal blood sampling and IUT before birth may be the better option and things could have changed, though no hard evidence supporting this concept is available right now. Immediately after birth newborns have to face an increased cardiac load associated with high energy used for adaptations to several changes, such as body temperature, fluid loss, and switching from placental respiration to pulmonary respiration etc. Accordingly, delivery itself can aggravate the compromised heart to fail.

## 4. Conclusions

The insights gained from this study are as follows: (1) Restrictive FO in a structurally normal heart can modify fetal response to anemia differently, by abnormally and unequally distributing blood volume, leading to much more deteriorating right ventricular function, meaning it is more vulnerable to develop failure. (2) To make decisions for IUT or EUT in cases of anemia-associated heart failure, several factors must be taken into account, such as gestational age, fetal cardiac function, and placental function. Because of the hyperdynamic state of newborns immediately after birth, delivery can deteriorate the compromised heart to irreversible failure. IUT for a well-prepared heart just before delivery may be the best option in cases of severe anemia in late preterm. The baby should be well oxygenated at the time of delivery.

## Figures and Tables

**Figure 1 diagnostics-12-01646-f001:**
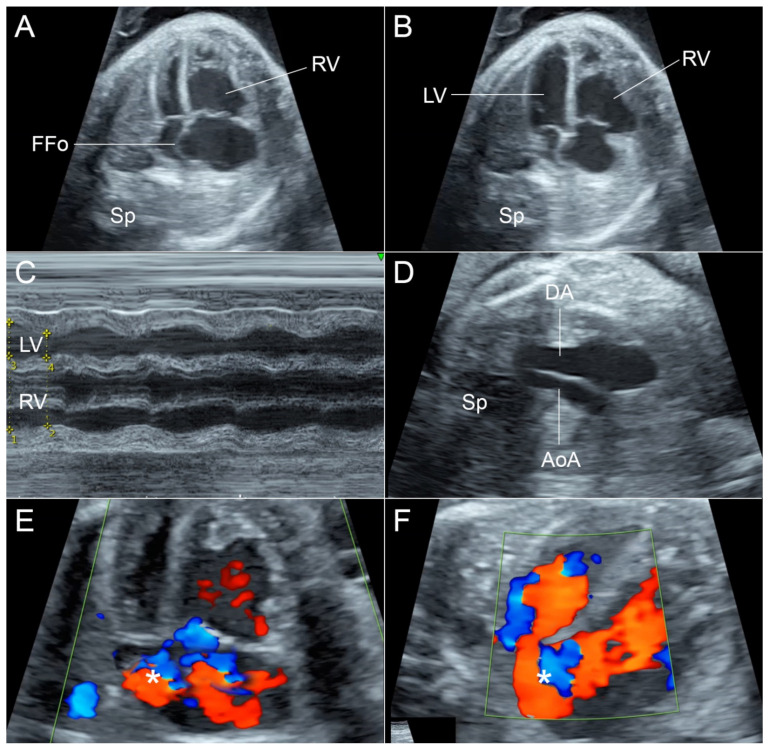
(**A**,**B**) Four-chamber views show left/right size disproportion and the size of RV is nearly the same during systole (**A**) and diastole (**B**), indicating poor contraction. The FFo is fixed and tense bulging during systole and diastole. (**C**) M-mode shows poor shortening fraction of RV, when compared with the LV. (**D**) Three-vessels and trachea view shows the much smaller AoA, compared with DA. (**E**,**F**) Color flow shows turbulent flow (*) in FFO and blood partially crosses to the left side (**F**). (AoA: aortic arch; DA: ductus arteriosus; FFo: flap of foramen ovale; LV: left ventricle; RV: right ventricle; Sp: spine).

**Figure 2 diagnostics-12-01646-f002:**
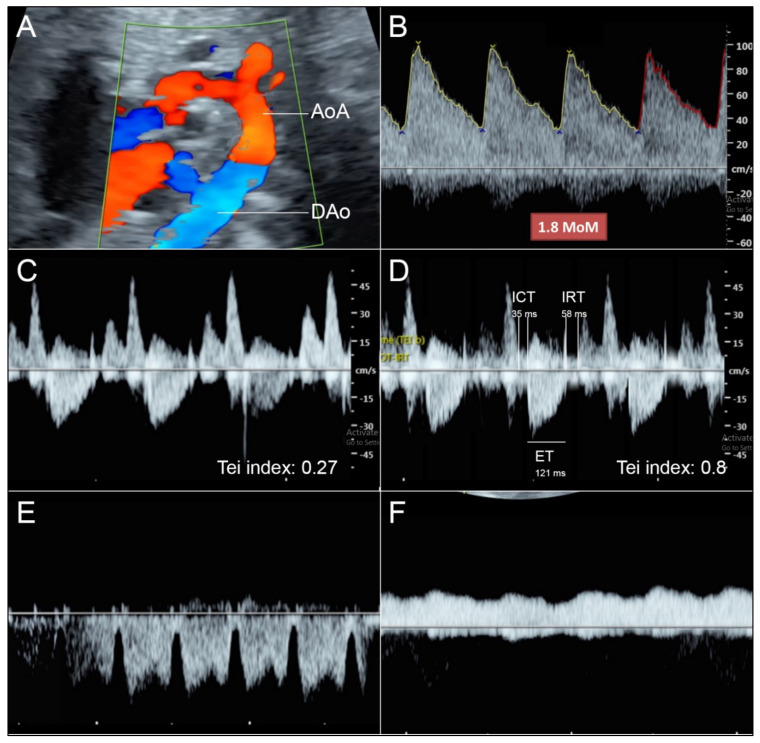
(**A**) Sagittal and cross-sectional scans of the arch view show reversed flow in AoA; (**B**) Markedly increased middle-cererbral artery peak systolic velocity; (**C**) Normal Tei index of the LV; (**D**) Markedly increased Tei index of the RV; (**E**) Ductus venous waveforms show very low a-wave, indicating increased preload; (**F**) Umbilical venous pulsations indicating high preload propagating cardiac pulse through ductus venosus. (AoA: aortic arch; DAo: descending aorta; FFo: flap of foramen ovale; LV: left ventricle; RV: right ventricle).

**Figure 3 diagnostics-12-01646-f003:**
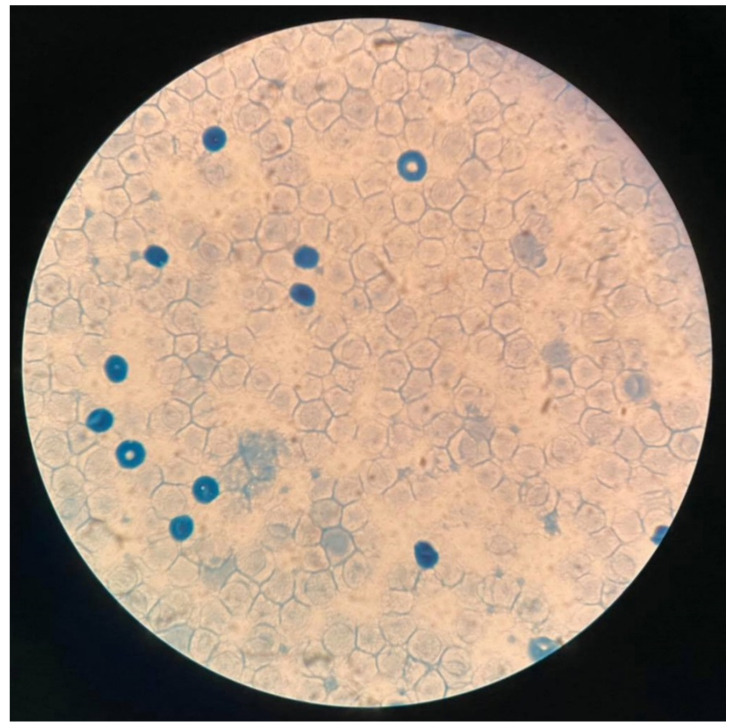
Acid-elution test shows numerous fetal cells (dark staining) in maternal circulation.

**Figure 4 diagnostics-12-01646-f004:**
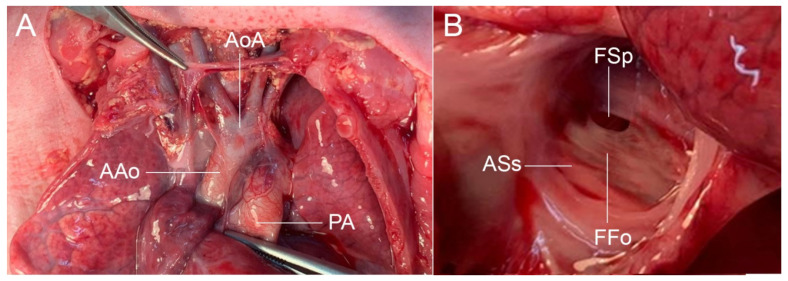
Autopsy findings: (**A**) small AAo and AoA, when compared with PA; (**B**) small FSp (AAo: ascending aorta; AoA: aortic arch; ASs: atrial septum secondum; FFo: flap of the foramen ovale; FSp: foramen of the septum primum; PA: pulmonary artery).

## Data Availability

The data of this report are available from the corresponding authors upon request.

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
