# Peer review of "The Accelerated Right Ventricular Failure in Fetal Anemia in the Presence of Restrictive Foramen Ovale"

_diagnostics, 2022, doi:10.3390/diagnostics12071646_

Round 1

Reviewer 1 Report

The author reported an interesting case of fetal anemia with accelerated right ventricular failure in the presence of restrictive FO.

I have a few questions and comments; I hope they can help and benefit the report.

  1. “….., since the restrictive FO of the structurally normal heart is very unlikely to cause heart failure [1].” Gu et al., in their publication “Isolated premature restriction or closure of foramen ovale in fetuses: Echocardiographic characteristics and outcome” reported right ventricular failure features with right atrial and ventricular dilation, tricuspid valve regurgitation, and effusion. Also, it is associated with increased complications and mortality.
  2. It would be interesting if you could report the size of the foramen in the echo and autopsy? Also, it will add value to your case if you can report some of the left side measurements from the autopsy examination to show the normal LV. 
  3. Continuation with the above point, I think all most likely be in the normal range. If that is the case, I would suggest revising the manuscript to discuss the functional obstruction that occurs across what would be anatomically normal FO due to hypervolemia in cases of fetal anemia. 
  4. “In this case, blood volume on the left side is much less than on the right side. Therefore, blood from the right side run-ning through the ductus and reversely runs in the aortic arch to feed the neck vessels.” Does the author think the flow reversal can happen due to volume change without a change in the pressure difference? If the LV function is normal according to your measurement, laminar flow through the aorta, and the placenta resistance is low, the aortic arch pressure should be higher, especially with reduced RV function. Do you have any supportive data or explanation for your statement?
  5. Ensure the abbreviations; in the abstract, the (FO) is written (Fo). “1) re-strictive Fo”.
  6. (or) repeated in 8th line in the introduction. “FO is usually redundant or or fixed bulging”
  7. I would suggest revising the title, as the main message from your case is the accelerated right ventricular failure in fetal anemia in the presence of restrictive FO

Thank you

Author Response

Reviewer: 1 

I have a few questions and comments; I hope they can help and benefit the report.

1) “….., since the restrictive FO of the structurally normal heart is very unlikely to cause heart failure [1].” Gu et al., in their publication “Isolated premature restriction or closure of foramen ovale in fetuses: Echocardiographic characteristics and outcome” reported right ventricular failure features with right atrial and ventricular dilation, tricuspid valve regurgitation, and effusion. Also, it is associated with increased complications and mortality.

Response: The phrase and citation (ref 12) is added, as highlighted in “Introduction” page 2.

2) It would be interesting if you could report the size of the foramen in the echo and autopsy? Also, it will add value to your case if you can report some of the left side measurements from the autopsy examination to show the normal LV.

Response: The size of the foramen is added in the second paragraph of “Case Presentation, and in the first paragraph page 4, as highligthed. However, we apologize that we cannot provide left side measurements because it is not recorded in the autopsy report.

3) Continuation with the above point, I think all most likely be in the normal range. If that is the case, I would suggest revising the manuscript to discuss the functional obstruction that occurs across what would be anatomically normal FO due to hypervolemia in cases of fetal anemia.

Response: As seen in Fig 4, the FO is obviously small, when compared to normal FO (5 mm at 35 weeks). In this case, the serious consequences were likely caused by the combination of both small FO and hypervolemia, not just functional obstruction.

4) “In this case, blood volume on the left side is much less than on the right side. Therefore, blood from the right side run-ning through the ductus and reversely runs in the aortic arch to feed the neck vessels.” Does the author think the flow reversal can happen due to volume change without a change in the pressure difference? If the LV function is normal according to your measurement, laminar flow through the aorta, and the placenta resistance is low, the aortic arch pressure should be higher, especially with reduced RV function. Do you have any supportive data or explanation for your statement?

Response: It is possible that both volume load and pressure load are responsible to the reverse flow in the aortic arch. However, based on evidence of flow, it seem to be mainly caused by volume load on the right side. In the revised MS, we tone down the comment to be “… probably caused by volume load rather than pressure load”, as highlighted at the end of the sixth paragraph, page 3.

5) Ensure the abbreviations; in the abstract, the (FO) is written (Fo). “1) re-strictive Fo”.

Response: In revised MS, restrictive Fo is changed to restrictive FO throughout the MS.

6) (or) repeated in 8th line in the introduction. “FO is usually redundant or or fixed bulging”

Response: In revised MS, this has been corrected, thank you very much for the observation.

7) I would suggest revising the title, as the main message from your case is the accelerated right ventricular failure in fetal anemia in the presence of restrictive FO

Response: The title is changed as suggested “The accelerated right ventricular failure in fetal anemia in the presence of restrictive foramen ovale”.

Reviewer 2 Report

This is an interesting case report explaining the role of restrictive FFO in determing a fetal distress. No clear relationship between anemia and restrictive FO can be found. I have just one question:

Is available a previuos fetal evaluation? No previous signs of this abnormality?

Minor issue: please check the paper for some typos 

Author Response

Reviewer: 2 

Comments and Suggestions for Authors

This is an interesting case report explaining the role of restrictive FFO in determing a fetal distress. No clear relationship between anemia and restrictive FO can be found. I have just one question:

Is available a previuos fetal evaluation? No previous signs of this abnormality?

Response: The fetus underwent ultrasound screening anomaly at 20 week (as mentioned in the second paragraph of  “Case Presentation”. Ultrasound was performed in other hospital, revealed normal. No abnormality was noted.

Minor issue: please check the paper for some typos

Response: The typographical errors are corrected.

Round 2

Reviewer 1 Report

Thank you for responding to my comment.

The responses were satisfactory except for comment 4.

The flow reversal in cylindrical tubes can not happen without differential pressure; volume can contribute to a change in the flow streaming only with the pressure differential between two sides.

In fetal anemia and anemia in the early neonatal period, aortic arch flow reversal phenomena have been described to be related to an increase in cerebral and coronary blood flow and vascularity due to low oxygen delivery. That leads to a reduction in the peripheral resistance in both organs due to autoregulation. As a result, pressure differences occur and reversal of flow. I think you already have a piece of evidence from increased middle-cerebral artery peak systolic velocity (Fig.2 B). 

Below are references hope they can help clarify the idea.

- Enhanced Coronary Blood Flow and Abnormal Blood Flow in the Aortic Isthmus in Severe Fetal Anemia. P Ramaswamy , G Greenstein, D Friedman, T Burgess, S Haberman. Pediatr Cardiol. Mar-Apr 2004;25(2):157-9. doi: 10.1007/s00246-003-0505-4.  Epub 2003 Dec 15.

- Acute and Chronic Fetal Anemia as a Result of Fetomaternal Hemorrhage.  Paul Singh and Tara Swanson, Case Rep Obstet Gynecol. 2014; 2014: 296463. Published online 2014 Apr 7. doi: 10.1155/2014/296463

Author Response

Reviewer: 1 (highlighted in red) Round 2

Comments and Suggestions for Authors

Thank you for responding to my comment.

The responses were satisfactory except for comment 4.

The flow reversal in cylindrical tubes can not happen without differential pressure; volume can contribute to a change in the flow streaming only with the pressure differential between two sides.

In fetal anemia and anemia in the early neonatal period, aortic arch flow reversal phenomena have been described to be related to an increase in cerebral and coronary blood flow and vascularity due to low oxygen delivery. That leads to a reduction in the peripheral resistance in both organs due to autoregulation. As a result, pressure differences occur and reversal of flow. I think you already have a piece of evidence from increased middle-cerebral artery peak systolic velocity (Fig.2 B).

Below are references hope they can help clarify the idea.

- Enhanced Coronary Blood Flow and Abnormal Blood Flow in the Aortic Isthmus in Severe Fetal Anemia. P Ramaswamy , G Greenstein, D Friedman, T Burgess, S Haberman. Pediatr Cardiol. Mar-Apr 2004;25(2):157-9. doi: 10.1007/s00246-003-0505-4.  Epub 2003 Dec 15.

- Acute and Chronic Fetal Anemia as a Result of Fetomaternal Hemorrhage.  Paul Singh and Tara Swanson, Case Rep Obstet Gynecol. 2014; 2014: 296463. Published online 2014 Apr 7. doi: 10.1155/2014/296463

Response: Thank you very much for your valuable comment. In the revised MS, we add the comment as suggested, as highlighted in page 3. Also we additionally cite the 2 references (Ref 18,19), suggested by the reviewers. Many thanks.

This manuscript is a resubmission of an earlier submission. The following is a list of the peer review reports and author responses from that submission.